# Towards Smart City Governance. Case Study: Improving the Interpretation of Quantitative Traffic Measurement Data through Citizen Participation

**DOI:** 10.3390/s21165321

**Published:** 2021-08-06

**Authors:** David Fonseca, Monica Sanchez-Sepulveda, Silvia Necchi, Enric Peña

**Affiliations:** Department of Architecture, Universitat Ramon Llull (URL), La Salle, 08022 Barcelona, Spain; silvia.necchi@salle.url.edu (S.N.); enric.pc@salle.url.edu (E.P.)

**Keywords:** urban sustainability, transport system, traffic sensors, traffic flow, participatory processes, quantitative and qualitative data, mixed method

## Abstract

Citizens play a core role in sustainable cities as users of the services delivered by cities and as active participants in initiatives aimed at making cities more sustainable. This paper considers the role of citizens as information providers and discusses the conditions under which citizens can participate in the development of sustainable cities. The objective of this study is to document the sustainability of an urban transit system and evaluate its compliance, with citizen participation as a major contributor. The methodology used is intensive field visits, interviews, and a mixed analysis of Sant Andreu de Palomar District in Barcelona city. The circulating vehicles are quantitatively monitored, qualitative problems are detected, and the typology of vehicles and other aspects identified and detailed in the study are indicated. All this information is contrasted with that of the technological sensors in the sectors. The results indicate that vehicles in the current pattern of urban density planned under incorrect sensor operation influence sustainable behavior through agglomerative clustering. This paper provides recommendations for future urban sustainability assessment research, including the employment of mixed-methods research, among other strategies. This article is intended to assist policymakers and traffic engineers in evaluating the sustainability of urban transportation infrastructure projects considering citizens as sensors.

## 1. Introduction

Sustainability in cities is a trend worldwide. As the world urbanizes, achieving sustainability in cities is quickly becoming a global concern [1,2]. Citizen participation is one of the most used terms in the sustainable city literature, showing that sustainable city initiatives need to create a community where citizens can engage more easily and effectively, thus developing citizens’ sense of ownership of their city, enhancing the local authorities’ awareness of their needs, and ultimately reshaping the citizen–government relationship [3]. Participation is a central concept in the public policy discourse and more generally in contemporary culture, and it has become one of the common threads of the digital age [4].

Citizen participation plays an important role in urbanism and a variety of mobility-related activities, including planning, policymaking, program and service design, and evaluation [5]. The involvement and participation of citizens in the sustainable urban mobility planning process is necessary to satisfy the needs of people and considerations that technological sensors do not detect. Citizen participation includes the valuing of non-expert or nonmainstream knowledge brought into the creative problem-solving development of planning [6].

Citizen participation is the act of generating new knowledge, adding new perspectives to the planning process, and disseminating knowledge to others in the process [6,7]. Van Herzle [6,8] found that the inclusion of non-expert knowledge was valuable in the planning process in general since the viewpoints of individuals external to the professional circle of urban planning can uncover creative solutions that could work in a specific local context. Corburn [6,9] stresses that local knowledge should never be ignored by planners seeking to improve the lives of communities, especially those facing the greatest risks. We are going through a period of some uncertainty about the urban reality and its evolution, with cities that, in their renewed territorial dimension, concentrate and intensify the problems and opportunities presented by the already evident changes of the time in which we are living. It is necessary to revise the urbanistic theories developed during the past decades that are adapted to the current reality [10].

The pacification of a city, understood as the processes that reduce traffic and make the coexistence of all the actors in the city more environmentally sustainable, is a priority for city councils and administrations at a global level. In the case of Barcelona, the aim is to advance towards the pacification of the urban space by focusing on having higher-quality air, more meeting spaces and fewer accidents and noise [11,12,13,14]. Decision making for action at any scale (streets, neighborhoods, or entire cities) requires not only detailed studies with sensors and direct observations but also the contextualization of the current society, the types of its interactions and the precise characterization of the particular and global needs of all its stakeholders. However, not all studies that propose feasible changes on a macro-scale are applicable to specific areas, and will depend in many cases on the design of each area [15].

Barcelona, as a typically urban case of study, is a useful or appropriate case study from the point of view of scientific approaches on how to make decisions towards a smart city of the future, especially in aspects involving urban mobility [16,17,18,19,20,21,22,23]. Barcelona’s location, limited by the Mediterranean Sea on the east, the mountains (Collserola Hills) on the west, the Besos River on the north, and the Llobregat River on the south, makes it impossible to grow, and amplifies the problem of traffic and urban mobility. Barcelona has a large central core with a grid design [24,25,26], surrounded by neighborhoods with old-village structures, assimilated by the growth of the city throughout the 20th century (Barceloneta, Sants, Gracia, Sant Andreu, etc.). 

In this sense, helping traffic measures also make it easier for citizens to opt for public transport and mobility on foot and by bicycle in their daily commutes. In addition to being sustainable, non-invasive, safe, efficient and accessible, greater mobility on foot makes it possible to fight against pollution and reclaim public space, which must once again belong to the neighborhood to improve quality of life. Due to its collective benefits and solidarity with the rest of society, mobility on foot, by public transport and by bicycle, is the mobility of the future [27].

Our study examines the nature and effects of participation in the planning of transit-oriented developments that are designed according to technological sensors. The aim is to validate the quantitative data provided by some different sensors through neighborhood visual qualitative samples, which can not only provide specific information that allows for a more sustainable redistribution of traffic, but also offers a low-cost solution for monitoring and characterizing traffic in specific urban areas with very specific mobility typologies. 

We document inner-city spatial transit mobility and its effects, as well as the significant effect of street networks on traffic performance in a larger area. By considering the role of citizens as sensors and information providers, this article will discuss how citizens can influence the success or failure of smart city initiatives by providing the information necessary for their implementation. 

Our approach is based on previous work that has shown how the creation of a participatory ecosystem in which citizens interact with public authorities is a key aspect for the design of spaces in a more satisfactory way [28]. This collaborative interaction leads to improved services as it is based on co-design and focuses on the user while identifying new governance models. Urban transformation in which citizens are the main “drivers of change” through their empowerment and motivation ensures that major city challenges, including sustainable behavioral transformations, can be addressed. Furthermore, these studies argue that city challenges can be addressed more effectively on a neighborhood scale and provide examples and experiences that demonstrate the feasibility, relevance and impact of such an approach, on which this study is based. 

The current study is carried out based on the following research questions:

RQ1: Is the pedagogical radar installed by the municipality correctly measuring the existing traffic volume?

RQ2: Given that an optical/laser or mechanical sensor does not pick up the type of vehicles circulating, what is the current typology of the traffic circulating on the pacified, pedestrian and non-commercial streets at present?

RQ3: In case of the restoration of the direction of Gran Street between Joan Torres and Malats, what would be the percentage reduction of commercial traffic in the pedestrian areas of the neighborhood?

RQ4: Why qualitative observations can improve quantitative sensor’s data?

In the next section, the objective is to identify the role of strategic traffic planning as a sustainable tool for developing an adequate and effective strategic model. In Section 3, the materials and methods used are described for a microscale traffic model of the traffic sensor data collected along the arterial historical center of Sant Andreu de Palomar in Barcelona city. In Section 4, we show and discuss the results of traffic data analysis, and we conclude in Section 5 with observations from our research considering the functions of economic, environmental, and social effects that increase the potential and benefit for a local area.

## 2. Context

At present, we can define cities as an entity with a life of its own and multiple agents that interact with this entity for its constant transformation. Undoubtedly, the citizen is the main transforming agent of the city, especially due to the use he/she makes of it and how he/she moves through it. 

The city as its own entity seeks its own sustainability and in a certain way its adaptation to all kinds of intelligent environments that are being implemented, while, on the other hand, its user, the citizen, seeks to improve his quality of life as an intrinsic aspect of the human being. That is, we could identify the need to establish a dialogue between the city and its inhabitants, a need that is established from the relationship between citizens and city administrations and the interaction with all kinds of mechanisms and technological systems of data exchange for decision making. 

Along with citizen participation, to optimize urban processes and improve citizens’ quality of life and the sustainability of the city, urban design requires resources and data to develop innovative, sustainable, and intelligent solutions. The concept of a ‘sensor city’ has appeared as an answer to the future challenges of growing urbanization and “datafication” [29,30]. Urban dashboards and platforms incorporating sensors [31], real-time monitoring stations [32], digital cameras [33], real-time tracking systems [34,35], big data analytical techniques [36,37,38], information and communication technology (ICT) [39,40], smart grids [41,42], and other digital appliances [43] with physical objects that indicate the urban context improve the efficiency of resource usage. 

These data contain visual, graphical and dynamic analyses able to holistically integrate, view and communicate real-time information on performance, trends, and future urban scenarios [44,45]. Their implementation is meant to facilitate the understanding of urban issues and provide a sense of accountability and engagement in smart urban governance activities.

Smart governance is based on principles such as transparency, accountability, collaboration (i.e., involvement of all stakeholders) and participation (i.e., citizen participation). Previous studies on the type of governance being implemented in smart cities [46,47] have identified that all initiatives rely heavily on technologies and there is some mix of collaborative, open and participatory governance. In this sense, it is clear to state that advanced technologies, innovation and smart governance are essential requirements for developing smart, creative, innovative and sustainable cities.

### 2.1. Urban Traffic Infrastructure 

Smart cities have the potential to make complete use of ICT to analyze, sense, and incorporate the principal systems of urban operations. This promotes the creation of a better urban life for citizens by making intelligent responses to factors such as people’s lives, public safety, environmental protection, and urban services [48,49]. The concept of the smart city, derived from the smart earth concept proposed in [49,50], consists of the insertion of sensors in every corner of a city, incorporating pertinent information through cloud computing and supercomputers. Smart cities promise to achieve a deep integration of informatization, industrialization, and urbanization, alleviate urban problems, strengthen urban management, and enhance the quality of urban life. Since the establishment of this concept, scholars have been exploring the theoretical and practical implications of the smart city [49]. Smart cities use advanced ICT to intelligently transform urban infrastructure to interconnect and integrate different parts of urban infrastructure and manage them more effectively [51].

Sustainability assessment and sustainability indicators are terms that have gained significant importance [52]. Noncompliance between sustainable development principles and transportation infrastructure is changing the appearance of historic cities from a livable and vibrant atmosphere into car-oriented conditions, causing environmental and social problems. Returning to pacifying the streets of a historical center when car culture has already been incorporated can bring difficulty in establishing a balance. 

Efficient methods and tools for road network planning and traffic management are critically important in the increasingly urbanized world [53]. The balance between the redistribution of traffic and internal mobility that is currently produced is necessary to achieve a sustainable urban development model. The growth of transport, which prioritizes cars, is associated with damaging effects on our environment, atmosphere and quality of life. To maintain the standard of transit that is required for society and the economy to function efficiently and not place pressure on the environment, it is necessary for governments to devise a policy that will take these factors into account. In this context, previous studies and research have identified different perspectives on motivating decision makers to balance citizens, industry and administration goals and wishes [54,55].

The main scientific approaches so far have focused on monitoring and counting traffic volume to obtain relationships between density and effects such as air and noise pollution. For this purpose, and with the objective of working as predictive systems of mobility and its associated variables (such as speed, vehicle flow or accident prevention), we can find proposals that make use of algorithms based on vehicle identification, deep learning algorithms, hybrid models or the combination of multiple data sources, such as loop detectors, travel time readings, and GPS location samples [56,57,58,59,60,61]. However, all these approaches have been used in macroscopic approximations of traffic (wide areas, intersections of large streets, highways, etc.), making an approximation by the typology of neighborhood, streets and traffic not approachable in our study as we have previously justified.

Focusing on the identification of the type of vehicle, a fundamental aspect in the motivation of our proposal, we found proposals that use magnetic sensor networks to measure traffic, occupancy, speed, as well as the type of vehicle [62]. Based on samples of 2 h measurements, it has been possible to identify more than 99% of vehicles, with a positive identification of the type of vehicles of about 60%, and with the potential to reach between 80 and 90% if algorithms based on the calculation of the wheelbase of the vehicles are applied. One of the limitations of this type of approach is that it is based on stable measurement points for both light conditions and measurement spaces, conditions that, in neighborhoods with narrow streets and high light variation depending on solar movement, are not applicable, in addition to the costs of implementation and configuration as other aspects that limit their use.

### 2.2. Citizens’ Participation in Urban Decision Making

People play a central role in cities as the direct or indirect beneficiaries of city activities. In addition to the view of citizens as passive recipients of the services delivered to them by the city, there is a different view when citizens assume an active role in the achievement of sustainable city objectives. The interests and concerns of citizens are coming to the forefront with the consciousness that a livable city not only consists of good infrastructure but also citizen input and feedback. The city as an objective reality and as a represented image plays an important role in the organization of space. However, the urban phenomenon that completes the urban structure is its representation as a social product, the result of human action [63]. 

Every human is able to act as an intelligent sensor equipped with a simple smartphone, a GPS (Geographic Positioning System)-based system, or even other means necessary to make measurements of environmental variables [64]. These voluntary data collection contributions have been standardized at all levels, especially those linked to GIS (Geographic Information Systems) applications, with data collection and mapping of all types of services, geographic features, sounds, pollution, valuations and even traffic-related aspects [65,66,67,68]. These approaches can enhance current e-government practices by enabling citizens to actively participate in urban decision making and service delivery, justifying the term “Citizen Science” which is often used to describe communities or networks of citizens who act as observers in certain domains of science [69]. 

Since sensor networks have not grown as fast as expected, and in addition to having a cost in which not all administrations are willing to invest, it is easy to find errors in their configuration and placement, making it necessary to find new alternative data sources. In the subjective field, we find studies assessing the capability of humans acting as a sensors, based on data collected from a wide range of social networks, the cell phone network or microblogs, to complement geo-sensor networks [70]. Current literature in the field of user-centric sensing often mixes different approaches on how data are generated, used and analyzed [71]. These measurements can be either subjective sensations, perceptions or personal observations, which identify the following terms:People as Sensors defines a measurement model, in which measurements are not only taken by calibrated hardware sensors, but in which also humans can contribute with their subjective “measurements”, such as their individual sensations, current perceptions or personal observations [71]. Measurements are not created in an absolute, reproducible way by calibrated sensors, but through personal and subjective observations. Such observations could be air quality impressions, street damages, traffic flow, weather observations, or statements on public safety.Collective Sensing does not exploit the measurements and data of a single person, but analyzes aggregated and anonymized data from collective sources, such as Twitter, Flickr, Instagram or the mobile phone network. This is a term related to People as Sensors, and we can find it as Participatory Sensing. Both terms, Participatory Sensing and People as Sensors, are very similar, but in this case, their definition is a little more restricted in terms of input devices, data acquisition and information processing [72].Citizen Science represents a human-based approach to science, in which citizens bring semi-expert knowledge and observations to specific research topics.

Previous works have studied how citizen networks have been initiated in the Netherlands, where official government control and business organizations leave gaps. The citizen-sensor networks formed collect information on issues that affect the local community in their quality of life, such as noise pollution and earthquakes induced by gas extraction [71]. These studies deal with monitoring systems that are established bottom-up, by citizens for citizens, to serve as “information power” in dialogue with government institutions. We find that these networks gain strength by joining efforts and activities in crowdsourcing data, providing objectified data or evidence of the situation on the ground on an issue of interest to the entire local community. By satisfying a local community need for information, a process of “collective sense-making” can grow combined with citizen empowerment, influencing social discourse and challenging the truth claims prevalent in public institutions.

Berntzen and Johannessen [73] introduced the term “citizens as sensors” as part of a discussion of the role of citizens in a smart city. Gil, Cortés-Cediel and Cantador [74] also discussed different forms of citizen participation in smart cities, with examples where citizens gather data to inform their government. Citizens as sensors encourage public participation in collective action, including taking inventories, making observations, and obtaining measurements that describe the urban environment [75]. Citizens can act as sensors by using their perceptions to detect and report problems [76]. Villatoro and Nin [77] refer to the citizen-sensor network as an interconnected network of people who actively observe, collect, analyze, report and disseminate information via different media such as text, audio, or video [78]. Citizens can participate in the development of smart cities and, as participants, can be (at least implicitly) considered responsible for this objective [4,79,80]. 

We consider citizens to play the role of qualitative sensors and information providers, following previous studies that have addressed this approach. For example, in [62], the authors present an urban traffic monitoring system based on participatory sensing, leaving the traffic detection tasks to the cell phones of bus users. Other approaches describe how citizens can serve as human sensors in providing supplementary, alternate, and complementary sources of information for smart cities, by collecting, communicating and sharing data with the corresponding administrations, through interviews, questionnaires, or simply using geo-tagged tweets [81,82]. 

In this sense, we can talk about the concept of Smart Communities with the idea of not just involving citizens in city governance, but rather to make them participate in the co-production of public services, and to help plan more effective engagement strategies for citizens, building users and stakeholders [83,84]. 

On the one hand, by having control over their user-generated data, citizens can contribute to public and nonpublic sectors to lead initiatives that can deliver public value. The example considered in this paper shows how looking at citizen participation in smart city initiatives through the lens of coproduction can allow cities to exploit the citizens’ active contributions to become smarter and can allow the public sector to be more productive and effective in creating public goods. However, this confronts the public sector with a serious challenge, forcing it to undergo a deep transformation to fully embrace the reality of participation and not only the rhetoric of participation.

### 2.3. Choosing an Approach: Qualitative Samples

In general, quantitative methods aim to provide summaries of data that help to validate generalizations about the studied phenomenon. To this end, they usually work with a small number of variables, automatic or semi-automatic collection, and rely on validation procedures to ensure the reliability of the results, which often requires statistical analysis. Quantitative results are limited because they provide numerical descriptions rather than details about the perceptions or motivations of the sample. 

Qualitative studies, on the other hand, begin with the broad goal of better understanding the human perspectives of the sample and end with a detailed description of a specific event or phenomenon of the experiment being evaluated. Their results are more difficult to compare, since their essence is precisely to obtain singularities in the responses. Thus, qualitative approaches make it possible to identify the main positive characteristics to be enhanced or the main weaknesses to be resolved in the sample studied. 

In the case of qualitative samples, the main criticisms are based on their “systematization”, or on the presumed lack of rigor derived from working with small samples, which is usually the result of a misinterpretation of the objectives of the study and/or of the responses obtained [85,86]. However, both quantitative and qualitative methods yield scientifically valid results. Their use and validity have been studied and compared in previous works and no significant differences in their scientific citation as valid supporting references are apparent [87]. 

For each qualitative method, Patton defines up to 10 types of characteristics with their function and the quantitative equivalent [85,88]. In ethnographic studies, the collection of observations, interviews and/or focus group work and organized documentary work are considered essential. These methods are particularly interesting and demonstrate their credibility [89], not only with small samples that can generate deviations in quantitative approaches, but also when, due to the typology of the users, it is necessary to obtain more personal information [90].

People play a central role in cities as the direct or indirect beneficiaries of city activities. In addition to the view of citizens as passive recipients of the services delivered to them by the city, there is a different view when citizens assume an active role in the achievement of sustainable city objectives. 

Observations from the affected population can be an important source of information due to their motivation. The volunteers in the study areas allow for an exhaustive collection of scientifically valuable data that provide explanatory insight to mechanically/automatically collected data. Data acquisition has always been a very expensive part of geospatial projects, and one of the areas where citizens can help. In this regard Goodchild (2007) coined the term Volunteered Geographic Information (VGI) [69], which conceptualizes the use of human power, knowledge, expertise and ubiquity in general in spatial data acquisition. VGI refers to “a series of geo-collaborative projects in which individuals voluntarily collect, maintain, and visualize information” [91]. 

VGI is also an implicit source of expertise of its users: the exploitation of information collected by a group of expert users in a specific spatial action can provide better solutions to other users, enabling them to perform that same action in the way an expert would [92]. If the effort to obtain these data goes from being a simple informative action to a service to support future actions, we find ourselves with the so-called voluntary geographic services (VGS), which use voluntary geographic information to aggregate, correlate and present information in a useful way for specific services [93].

Citizen involvement in social changes is often valuable in its own right because it makes the development of solutions to societal problems more transparent to the people affected by them. There are also other common reasons for citizen engagement: citizens bring local knowledge about the problem and their needs; they can generate new solutions informed by their knowledge; and they bring different points of view, leading to more diverse perspectives on the problem. When involved in the process, citizens are also more likely to accept the solutions [93].

## 3. Materials and Methods

Active design feedback from citizens is a fundamental component in making a responsive city. Therefore, we argue that implementing sensor networks is a necessary but not sufficient approach for improving urban living [94]. 

This research focuses on citizens as an active member to collect qualitative data for urban decisions. The active position of the citizen helps in collecting data about their surroundings; they observe and report [76]. Information obtained from citizens as passive entities helps in understanding and optimizing smart city functions, and citizens as active entities motivated by their common sense communicate the sensed samples.

The present study is developed in the area of Sant Andreu del Palomar, which is the oldest part of the Sant Andreu district in Barcelona (see Figure 1). This district remains compact, with more than 57,961 inhabitants [95]. Although the extension of the district is larger, the nucleus or old town is small, with narrow streets, private homes, and some basic commercial sites. 

The problem of the studied case starts with the pacification and change of the traffic direction of the Gran de Sant Andreu Street (as we can see in detail in the Method section). With the current change in the direction of traffic on the main street, there has been a notable increase in traffic on the surrounding streets. This increase is critical, since these streets are mostly single-platform streets previously pacified, narrower than the main street, with no commercial uses and mostly used mainly for housing and private homes. As a result of this situation, the neighborhood platform FemCrida (@Transitstandreu) was formed in 2019, which conveyed to the Barcelona City Council the need to institute a distribution of traffic that truly pacifies the streets of the neighborhood equally.

This district remains compact, with more than 57,961 inhabitants [95]. Although the extension of the district is larger, the nucleus or old town is small, with narrow streets, private homes, and some basic commercial sites. The problem of the studied case starts with the pacification and change of the traffic direction of the Gran de Sant Andreu Street (as we can see in detail in the Method section). 

With the current change in the direction of traffic on the main street, there has been a notable increase in traffic on the surrounding streets. This increase is critical, since these streets are mostly single-platform streets previously pacified, narrower than the main street, with no commercial uses and mostly used mainly for housing and private homes. As a result of this situation, the neighborhood platform FemCrida (@Transitstandreu) was formed in 2019, which conveyed to the Barcelona City Council the need to institute a distribution of traffic that truly pacifies the streets of the neighborhood equally.

Sant Andreu del Palomar is experiencing a new episode in the debate on road traffic on Gran Street, the neighborhood’s historic artery and commercial epicenter. This main axis of the town has changed its traffic direction, now taking cars, delivery trucks and buses that circulate in the surrounding lanes in the direction of the city center, which creates inconvenience in part of the neighborhood. In the past, the town of Sant Andreu was characterized by being a small town where proximity was the pattern that governed daily life. One could walk to the main points of the city, such as the town hall, church, artisans, shops, services, fields and other places of work [96]. 

This means that work, recreation, meals, discussions and social contacts were interlaced. Once the town started to grow, so did its infrastructure. We note that the historic centers that have managed to preserve their old and renovated commercial infrastructure, as is the case in Sant Andreu, are important driving forces of urban life. The question is how to reduce traffic on this street without completely blocking it, since there are many private garages in the area and stores need to carry out loading and unloading tasks every day.

The present study is based on the data provided by the Barcelona City Council collected by the pedagogical radar installed to justify the change of directions in the streets of the Sant Andreu neighborhood. Because the data supplied to the neighbors did not correspond to their perception, a qualitative sample collection was designed under the following premises:Given the limited technical, temporal and economic possibilities of the platform, we opted for a qualitative approach (averaged individual samples). Given that different types of vehicles (taking into account business hours) can be found in terms of sample collection schedules, this is a stratified sample within the typology of mixed studies defined in [97].These samples were designed to capture 10 m periods of traffic in order to complete 2 h cycles throughout different daily slots, both on weekdays and weekends. The objective of this process was to start from a common basis of comparison between the different measurement points with a significant sample, following the example of [62].There are different approaches that can guide the design of sampling strategies for open space or environmental monitoring, based on the use of different numbers of individual or pooled samples. Previous work [98], shows that these are appropriate depending on cost and sample variance. However, when analyzing new sites, the recommended approach is to initially use individual samples in time series, which allows for better refinement and detection of sample variance.Four measurement points, one at the entrance to the conflict zone (P1), another after the first possible detour (P2), the third at one of the most complicated streets and turns technically prohibited for commercial vehicles and which is also an exit point of this new route (P3) and finally one at the exit of the study area (P4).

Although the initial objective was to monitor and compare the diversity of traffic throughout the day at the measurement points, thanks to the data provided by the Barcelona City Council, the study goes to a higher level, positioning itself as a complementary tool to the mechanical quantitative approaches carried out by the City Council. As we will see later, a data sharing process has been initiated between the City Council and the platform that exemplifies how citizen collaboration and participation (smart governance) improves smart cities and the technological processes applied to its monitoring.

### 3.1. Location, Motivation and Project Description: The Problem

The main nucleus of the old town can be defined by the limits that form Palomar Street in the north; in the east, the axis is formed by the Streets of Torres i Bages and Segre, in the south by the Ramblas de Once de Setembre and Fabra i Puig, and in the west by Meridiana Avenue, as seen in Figure 2. This forms an area of approximately 826,983 m^2^ and a perimeter of 4170 m.

The pacification of the main street (shaded and identified by areas A, B, C, and D in Figure 2; also known as Gran Street) and the change of direction in the cycle (now the N direction) from Malats Street to Palomar Street (historically, it was completely in the S direction, like the rest of the street) constitute the starting point of the problem that affects other streets. The new direction of traffic by street is shown in Figure 3.

The change in direction described notably affected the internal traffic of the neighborhood, since the vast majority of vehicles that come from northern Barcelona and its ring roads pass through Input Point 1 (i1), which is adjacent to Sant Andreu of the Palomar neighborhood. The other main entry points to the neighborhood are identified as i2, i3, and i4 (access from the city center) as well as i5 and i6 (east access, from the coastal area).

Initially, the configuration of the streets (as we can see in Figure 4), made it possible to reach the most commercial area of the neighborhood, zones B-C-D, by entering from zone A (red dashed line of Figure 4), and driving south, or by entering from point 1 and turning onto zone B (black dashed line of Figure 4). 

The new configuration of Gran Street, after the pacification by the City Council, is shown in Figure 5.

As seen from Figure 5, vehicles that enter through i1 (Joan Torras Street) to reach the center of the neighborhood (zones B, C and D), which has the highest commercial and residential density, cannot turn along Gran Street (zone B, with a length of 190 m) and must travel through pedestrian streets and previously pacified areas, such as Agustí i Milà, Segarra, Marià Brossa, Matagalls, and Malats, and then turn onto Gran Street (to complete a path of approximately 500 m and rejoin the target route; see the dashed black line in Figure 5). In addition, there is a growing volume of vehicles that use the bypass by Servet i Arbúcies to access the central area of the neighborhood, all the roads of which are narrow streets, with closed and previously pacified turns. On the other hand, part of the traffic that could previously access the neighborhood by driving south along Gran Street (dashed red line in Figure 4), now has to go around the neighborhood and enter through point i1, which concentrates all the entrances from the east of the neighborhood.

The motivation for the study starts with the notable increase in traffic that is generated at Input Points i1 and i2 to move towards the center of the neighborhood (private vehicles) and access commercial areas (A-B-C-D, for commercial vehicles). This volume of traffic has generated an increase in pollution, noise and vibrations on the affected streets, breaking urban furniture, pavement, and trees and even causing accidents.

In Figure 6, we can observe the main points measured, taking into account the new routes of the traffic identified in Figure 5: P1–4: Qualitative points of observation.P2: Initial location of a pedagogical radar (quantitative data).M1–5: Quantitative data recollected by mechanical sensors.N1–4: Non-measured critical points.

### 3.2. Initial Data: Optical Sensor of Pedagogical Radar

In the first instance, and after more than a year of complaints and neighborhood actions, the Barcelona City Council installed informational radar on Segarra Street (a single-level street with pylons and a maximum speed of 20 km/h; characteristics of the sensor: https://www.radarespedagogicos.com/mini-sp, accessed on: 14 March 2021), finding that the flow of vehicles was approximately 400 a day, which was considered low traffic (see Figure 7).

These data presented to the platform in early 2021 were based on two-week samples with a data summary as shown below:Week of 15–19 February 2021: Total vehicles: 1676.
With a measured average speed of 21 km/h and a maximum of 62 km/h.Percentage of speed violations: 50.84%.There are no samples from Friday, 19 February 2021, to Sunday, 19 February 2021.The counts of vehicles on the four business days are:
○Monday 15: 394; ○Tuesday 16: 423; ○Wednesday 17: 430;○Thursday 18: 428.
Week of 20–26 February 2021: Total vehicles: 2249.
With an average speed of 21 km/h and a maximum of 53 km/h.Percentage of speed violations: 48.92%.There are no samples from Friday, 24 February 2021.The counts of vehicles on the four monitored working days are:
○Monday 20: 396; ○Tuesday 21: 460; ○Wednesday 22: 453; ○Thursday 23: 458; ○Saturday 25: 276; ○Sunday 26: 202.

### 3.3. Method

Based on the data received from the first sensor installed, and since the data were not perceived as real, a qualitative study was designed at 4 key points in the conflict zone. In order to validate or not the data collected by the optical/laser sensor installed by the Barcelona City Council, and given that it does not collect the type of vehicles circulating, the qualitative data collection presented in the article was designed. In particular, the data referring to the type of vehicles circulating on the streets studied are a fundamental aspect to understand the problems involved in the pacification of a single street instead of an entire environment. The collection of samples from optical sensors such as the one installed, or the mechanical ones that were installed a posteriori and that we will analyze later, does not collect the typology of vehicles and their relationship with the environment, a fundamental aspect that the qualitative approach aims to demonstrate as an approach that allows for a more correct form of decision making and is adjusted to the reality of the environment. 

The qualitative visual study was carried out throughout the month of March 2021 at the following four strategic points (see Figure 6):P1: Agustí i Milà Street at number 9 (beginning of the street, with partial visualization of vehicles with problems turning towards Segarra Street). This is the entry point of the new loop derived from the change of direction of the main street, being a relevant point to measure.P2: Segarra Street at number 6 (visualizing the vehicles that turn from Agustí i Milà towards Segarra, those that make the prohibited turn and circulate in the opposite direction from Tramuntana Street, those that continue along Agustí i Milà towards Meridiana Avenue, and the speed captured by the radar in its initial position as well as possible measurement errors).P3: Matagalls Street with Malats (continuation of the traffic that enters through Segarra Street and goes to the center of Sant Andreu). This is the exit point of the loop.P4: Servet Street with Arbúcies (vehicles that go up Servet towards Meridiana and/or turn onto Arbúcies Street, making a closed and narrow turn forbidden to trucks, towards the area of Sant Andreu—Fabra i Puig Avenue). This is a secondary exit point of the new loop, in a street with the type of pavement and configuration clearly focused on pedestrian mode.

Samples were taken at 10 min intervals at each of the four measurement points, and they were divided into three large time bands, morning (07:00 a.m.–1:00 p.m.), noon (1:00 p.m.–4:00 p.m.) and afternoon (4: 00 p.m.–9:00 p.m.), separating business days from holidays. Additionally, it should be noted that the measurement period was conditioned by the circumstances derived from local, provincial and national actions to regulate and contain the COVID-19 pandemic. Therefore, the monitored period was subject to the following restrictions: (a) There are time limits for bars and restaurants in the band from 7:30 a.m. to 5:00 p.m. Pickup hours are allowed from 7:00 p.m. to 10:00 p.m. (b) Shops open during permitted hours (latest closing at 9:00 p.m.) and are closed on weekends if considered nonessential. (c) A curfew restricts going out from 10:00 p.m. to 6:00 a.m. These restrictions were able to influence and alter the samples that were collected, both by the systems and sensors installed by the Barcelona City Council and by the visual sampling conducted in the study. In this sense, it is understood that, with an unrestricted opening, the traffic would be greater and would extend to new unmonitored schedules that, in any case, would only increase the volume of registered traffic.

This qualitative sample collected visually monitored the type of vehicles circulating at each point. This information is relevant as long as the damage to the pavement, bollards, trees and other elements of urban furniture that the area suffers is justified. These data are not collected by either of the two types of sensors installed by the City Council (the informational radar in Segarra Street (laser technology) uses the K-band frequency (24 GHz), https://www.radarespedagogicos.com/mini-sp accessed on: 14 March 2021) or by the five mechanical circulation counting points that were installed during the week of 9 March 2021 to 16 March 2021 (see Figure 8).

The latter sensors are cables called “pneumatic tubes”, and they are used throughout the world to analyze the traffic that circulates on a road or street. These cables are hollow, and air circulates through them. When a vehicle passes over them, the air is compressed and sent to both sides of the cable. At one end is a small mechanical sensor, a “switch” that sends an electrical signal to a counting system, usually articulated in the form of simple software. In addition to counting the number of vehicles in a time frame, it can count the time interval between vehicles. The results of the collection of these samples will also be analyzed in the next section.

## 4. Results

### 4.1. P1: Main Sample

This was identified as a problematic entry point (i1, Joan Torres Street), and given the current impossibility of turning on Gran Street, at P1 (collection point 1), the number of vehicles that entered, their types and those that caused turning maneuvers due to their size when accessing Segarra Street were monitored (change of direction problems, CDPs; see Figure 9).

A total of 54 samples at 10 min intervals were collected, equivalent to 9 h of actual circulation, covering the three previously defined daily slots and differentiating Saturdays and Sundays. For each period, the types of vehicles that circulated were recorded separately:V1: motorcycles;V2: private cars (without commercial signs);V3: small commercial vans of car size (officially known as “tourism derivatives” with the code 30** on their vehicle registration certificate, according to https://campermania.es/clasificacion-vehiculos/, accessed on: 14 March 2021);V4: large commercial vans or any type of truck (including 31** adaptable mixed vehicles, 24** vans, 10** passenger cars, and any other type of delivery truck with a length of more than 5 m).

A probabilistic sample (random, with the same options for any type of vehicle) was obtained with a target universe of 3000 vehicles, defining a 95% confidence interval. Therefore, for the 2758 vehicles monitored, the estimated error was 0.53%. The collection of samples began on 5 March 2021 and ended on 15 March 2021. Table 1 shows the data collected.

The average number of vehicles per hour that circulate through the entrance of Agustí i Milà Street in the morning is approximately 362, with the majority being private vehicles (V2: 48.07%), followed by commercial vehicles (V3 + V4 = 35.92%) and finally motorcycles. In particular, the percentage of large vehicles (V4 = 20.86%) stands out, of which approximately 50% turn on Segarra Street (CDP = 9.25%), with a resulting problem when turning. This volume of vehicles is significant, given that neither Agustí i Milà Street nor Segarra Street and the rest of the circuit currently traveled by vehicles have commercial loading/unloading activities, so it is assumed that these are either vehicles leaving the neighborhood (going towards Meridiana Ave.) or vehicles forced to use this route because of the current circulatory arrangement.

The standard deviations below 25% in the 10 m strips for the V3 and V4 vehicles show the same constant circulation throughout the entire schedule defined in the morning sample, converging at school entrances, which generates a large pedestrian volume due to the presence in the same Agustí i Milà Street of a kindergarten and primary, secondary and high schools.

If the progression of the average number of vehicles in this strip were extrapolated to a working transit time of 6:00 a.m. to 10:00 p.m., 16 h (it should be remembered that at the time of the sample, the curfew was being implemented from 10:00 p.m. to 6:00 a.m.), it would yield a daily circulation of 5792 vehicles, with the hypothesis that between 40 and 70% of the vehicles would turn through Segarra Street due to the current arrangement. This hypothesis opens a range of between 2316 and 4054 vehicles circulating along the Segarra–Matagalls or Segarra–Servet axis, all due to the impossibility of access to the central area of the neighborhood.

In Table 2, we can see the breakdown of the samples collected in the second band of the working day (from 1:00 p.m. to 4:00 p.m.).

The average circulation per hour is maintained with respect to the first strip, with a total of 368 vehicles, of which 56.8% (an increase of approximately 10% compared to the previous strip) are private vehicles, and the circulation of commercial vehicles (V3 + V4) is reduced to 25.8%, followed by a greater reduction in large vehicles, matching business closing hours.

Analyzing the data in 10 m strips, a low SD is observed globally (of 7.09 out of 61.33), which indicates a constant flow of vehicles at times that coincide with school letting out at 1:00 p.m. and beginning again at 3:00 p.m., causing greater traffic of children in the street, with consequent danger due to traffic circulation. We could conclude that the progression of daily traffic would be maintained with respect to the first strip studied, placing the traffic above 5000 vehicles per day.

To complete the study of traffic on business days in P1, Table 3 shows the samples collected in the range of 4:00 p.m. to 9:00 p.m.):

The evening average rises to 416.5 vehicles per hour, an increase of 15%. The average number of private vehicles rises (exceeding 65%), and commercial traffic drops to 15% (V3 and V4). Approximately 50% of the V4 vehicles continue to turn with difficulty and with the need for maneuvers. This percentage is maintained throughout the day. The daily progression of circulation considering the afternoon–nighttime band would rise above 6000 vehicles per day, which could be defined as the upper margin of global circulation over 24 h.

If we globally analyze the traffic registered in P1, the average on a working day is 385 vehicles per hour: 70 of type V1, 219.6 of type V2, 42.2 of type V3 and 53.2 of type V4. The V1-type vehicles constitute 18%, the private ones (V2) are 57% and the total of V3 and V4 is 25%, reflecting turning problems in 7% of the vehicles (50% of the V4 type).

The preliminary results for measurement point P1 show a prediction of traffic of 6.00 to 22.00 of 6160 vehicles, which turn by Segarra Street between 40 and 70% of the time (2464–4312 vehicles) according to the time of day. As seen in Figure 10, of the 30 samples collected on a working day (each one of 10 min), the minimum value of traffic does not fall below 40 vehicles in 10 min (for an average of 240 vehicles/hour and 3840 daily), with a maximum of 81 and an average value of 64.16 vehicles every 10 min.

Table 4 and Table 5 below show the data collected in P1 on Saturdays and Sundays, respectively. On Saturdays, as seen in Table 4, traffic decreases by approximately 30%, an aspect that may also be influenced by the restricted hours of commercial activity due to the rules dictated by the COVID-19 pandemic (closure of nonessential stores and early closure at “16.00” for restaurants).

During business hours on Saturday mornings, the frequency is approximately 50 vehicles every 10 min, dropping in the afternoon (14–18) to approximately 25–30 vehicles and rising again to 50 vehicles (with a majority being private cars) in the period from 18 to 21 h. Due to these variations, the standard deviation of the mean is very high and shows a large fluctuation between the samples taken. Subjectively, the drop in traffic is perceived as an increase in speed, especially for motorcycles, which is reflected in greater danger on the street. For commercial vehicles, the data obtained on working days are maintained, with 50% reporting turning problems, reaching a percentage close to 15% of the overall figure. The daily progression with respect to the average is established at 665 vehicles per day. On Sundays, traffic is reduced to its weekly minimum, as shown in Table 5.

Commercial vehicles (V3 and V4) account for 10% of the recorded traffic, while the problems for 50% of the V4 vehicles remain approximately unchanged and private vehicle traffic reaches its peak (approximately 75%). As in the case of Saturdays, there is an increase in the standard deviation, which reflects a clear differentiation between time slots: a higher traffic volume in the early and late hours of Saturday and late hours of Sunday compared to the rest of the time slots. The daily progression would be approximately 2672 vehicles. 

The overall summary reflects a much higher traffic volume than that presented by the Barcelona City Council on the basis of the data collected by the informational radar. Considering that the citizen samples analyzed do not reflect the traffic at the exact point where the radar was placed, the samples were collected at the complementary points described below.

### 4.2. P2: Secondary Sample: Segarra Street

Given the degree of uncertainty in the number of vehicles turning onto Segarra Street according to sample P1, a complementary sample was taken at P2. The sample consisted of accurately monitoring the following:The number of vehicles that continue along Agustí i Milà Street or turn onto Segarra Street;Of those that turn, checking those with turning problems and the need to maneuver;Checking the speed at which the training radar captures them for the first time;Checking whether there are vehicles that, due to masking by other vehicles and/or uncontrolled situations, do not reflect the passage of vehicles;Finally, detecting the vehicles that join Segarra Street from Tramuntana Street, circulating several meters against the direction of Agustí i Milà Street.

This control point serves as a link between the data collected in P1 and those that we will present below for points P3 and P4. The sample was taken in five 10 min strips (two in the morning, one at noon and two in the afternoon), maintaining equity with the sample at P1. The data obtained can be seen in Table 6 and in the summary in Table 7.

Relating these results to those of P1, we can make the following observations:The average number of vehicles per hour in P1 is 385, with 18% of type V1, 57% individuals (V2) and 25% for the sum of V3 and V4.The average number of vehicles per hour in P2, focusing on Agustí i Milà Street, is 338.4 vehicles per hour, of which 219.6 (64.9%) turn onto Segarra Street and the other 118.8 (35.1%) continue along Agustí i Milà Street. This amount is within the 10% tolerance of the observed morning and midday flows and approximately 15% of the observed afternoon flows in P1, which is consistent with the measures taken.The percentage distribution by type of vehicle is practically the same in P2 as in P1 in the mornings, with approximately 50% of private vehicles, followed by 35% of commercial vehicles and approximately 15% of motorcycles.12% of the vehicles require maneuvering and/or encroachment on a sidewalk, tree-lined area, or pylon area. This accounts for 66% of V4-type commercial vehicles.

The data collected at this point are very relevant since they directly contradict the data captured by the informational radar sensor that were initially presented by the Barcelona City Council for the same position (RQ1). In contrast to the initial sensor data that collected averages of at most 35 vehicles per hour, the on-site sample generated averages of 210 vehicles per hour (600% of the data collected by the informational radar). On the other hand, as seen from Table 7, up to 15% error is identified in the vehicles that the radar sensor identified (based on its activation). This error is random and was found to affect the detection of motorcycles as well as cars and vans, circulating both in isolation and in rows—that is, in cases of isolated circulation, the sensor is not activated, and likewise, in grouped rows of vehicles, there is a masking of subsequent vehicles, especially if the first ones in the row are large.

These data are especially relevant since much of the reasoning of the institutions for the current map of street directions is based on data collected automatically, which showed a null incidence in the neighborhood spaces of such a change in traffic. The disparity of the measurements may be due either to a bad calibration of the sensor or to a bad placement of the sensor that caused the lack of sensitivity identified. Other complementary but relevant data at this measurement point were collected as follows:A total of 33% of the vehicles are speeding. Depending on the initial sensor sample, this percentage can be as high as 50%.As shown in Table 7, practically all types of vehicles on average exceed the maximum speed allowed at the point of measurement, with motorcycles (V1) and small commercial vehicles (V3) standing out. These data should be interpreted from the perspective that the sensor identifies them when accelerating, which predicts that the passage through the area of Marià Brossa Square is still at a higher speed.Seven percent of the vehicles counted in P2 come from Tramuntana Street, performing an illegal maneuver with circulation in the opposite direction on Agustí i Milà Street.

### 4.3. P3: Secondary Sample: Matagalls Street (Starting in Segarra) with Malats 

The sample at this point was collected in different time slots from 8 March to 23 March 2021. This point is fed by vehicles that enter from Segarra Street coming from Agustí i Milà street, those that have gone down Sant Hipòlit Street and turned into Marià Brossa Square, and finally vehicles circulating in an ascending direction by Servet Street and turning by Matagalls Street. The summary of the samples is shown in Table 8.

As seen, feedback on this axis is identified with the traffic already observed in P1 and P2, and now the flow coming from the two entry points to this sector is identified. From a percentage point of view, 93% of the vehicles entering from Segarra Street arrive at this intersection, 7% of which enter the only parking lot on Segarra Street (with approximately 250 parking spaces) and/or circulate westbound on Servet Street. Likewise, the average collected vehicles per hour would be 61% of the traffic that circulates in Agustí i Milà Street, demonstrating the overload of this street by the traffic towards the center of the neighborhood.

At this point of measurement, as at the previous one, no distinction is made between those vehicles circulating in an easterly direction (going down Sant Hipòlit Street), those coming out of the parking lot on Segarra Street, and those coming from Servet Street in the westward circulation. Later, with the data provided by the Barcelona City Council collected by the mechanical sensors, we compare the results to establish the behavior of traffic on this alternative route to the center of the neighborhood.

### 4.4. P4: Secondary Sample: C/Servet Corner C/Arbúcies

As we will see below, given that the measurement by means of mechanical sensors by the Barcelona City Council did not consider the turn from Segarra Street to Servet Street, the platform decided to set a sample point on this street at the junction with Arbúcies Street, a very tight, narrow and complicated turn even for small vehicles, which has a turn prohibition sign for V4 vehicles. The possible flow of vehicles has two origins: those from Segarra Street turning towards Servet Street, and those that already circulated from the lower part of Servet and continue along this street towards Meridiana Avenue. The procedure used was the same as that used for P1 (in this case, based on video recording and subsequent calculation), separating the samples into the three identified daily strips (see Table 9, Table 10 and Table 11).

The average throughout the day remains constant at approximately 14–16 vehicles every 10 min for an overall hourly rate of approximately 90, as shown in Table 12. This volume is especially important given the type of street (cobblestone paving) and the problematic turn with Arbúcies Street. Given its paving and speed limitations, the reflected traffic volume generates high noise pollution. In addition, in the monitored strips, 47% of the vehicles make the turn with Segarra Street, constantly invading the space of a tree that currently threatens to fall on top of a house next to it due to the constant impacts of vehicles.

### 4.5. Additional Sensor Data from Mechanical Circulation Counting

As indicated above, in view of the initial data from the speed sensor, which contradict the results of the qualitative monitoring explained in the previous four sections, the Barcelona City Council installed five measurement points (M1–5) for one week, as shown in Figure 6. The analysis of the results was as follows:M1: This sensor captured an average of 4985 vehicles per day over a week. This is considered to be 100% of the entries in the study area.M2: At the second measurement point, the sensor picked up an average of 3537 vehicles per day. The interpretation is that 1448 vehicles continued along Agustí i Milà Street (the point referenced as N1), so 70% of the vehicles turned onto Segarra Street.M3: At the third measurement point, a total of 3052 vehicles were collected, which, according to the Barcelona City Council, can be interpreted as 60% of the initial vehicles. It is assumed that only 485 vehicles traveled along Servet Street (indicated on the map in Figure 6 as point N2).M4: Given that 772 vehicles were collected at this point, it is assumed that they were only 15% of the initial vehicles and that 45% circulated along Malats Street at point N4 (2280 vehicles) and only 259 vehicles entered through N3.M5: A total of 2539 vehicles were registered at the last measurement point.

This analysis of the data collected by the mechanical sensors clearly shows several defects of form that increase when moving through the study area. In summary, the data to be reviewed and/or considered should account for the following:Point M1 collects the entrance circulation to the study area data, which are global data that are not debatable.Point M2 is not fed only by the traffic of point M1, since it considers the vehicles arriving at this point from Tramuntana Street (with traffic violations) and those coming from Sant Hipòlit Street.Given the sample of M3, it is subtracted from M2 to conclude that only 485 vehicles circulate through the point indicated as N2, which is false, since those coming from Zone B, in ascending circulation through Servet Street, are not considered.Sample M4 is not fed exclusively by M3 since Servet Street is an important entry point to the neighborhood, marked on our map as i2. It should be noted that no sensor was installed on this whole axis of Malats Street, which has much traffic that not only circulates in this street but also comes from and feeds into side streets. In summary, the percentages and absolute numbers of points N3 and N4 are not correct, since they are obtained from the subtractions of M5, M4 and M3, without considering the inflows of up to three points, one of which is the main one, i2.A sample of the error generated by the interpretation of the results can be seen in the data collected at M5: 2539 vehicles. This figure is higher than the 2280 predicted as 45% of those entering from M1 according to the study and marked at point N4. Given that the only way to access M5 is to circulate through N4 and that vehicles circulating through N4 can be diverted through two streets (both exiting the neighborhood) without passing through M5, the value at N4 should be higher than that at M5, invalidating the percentages and calculations of the traffic between M2 and M5.

## 5. Discussion

In the particular case studied, it has been shown how the pacification of an entire commercial axis through the paving of a single platform, modifying certain traffic directions to favor pedestrian traffic, has caused the de-pacification of several previously pedestrian, residential streets with a low volume of vehicles.

Political intentionality in this type of decision is avoided beyond decision making that may result, as the study has shown, from the data of poorly placed or calibrated sensors. If the objective is to reduce traffic on an axis, this has to be approached globally by aiming to reduce traffic on adjacent roads and preventing vehicles from seeking alternative routes to reach their destinations. In other words, pacifying an axis requires pacifying the entire environment and preserving the overall sustainability of the entire urban environment. This process, in itself, is currently complicated and close to impossible to carry out, since the volume of traffic in cities is constantly growing as cities themselves grow in size and services. At present, door-to-door services, commercial deliveries, loading and unloading, and private traffic are on the rise so that any tactical decision for a street or a specific axis can only lead to the collapse of the surrounding areas.

The present study, as well as that related with the RQ1, has demonstrated that the pedagogical radar installed by the municipality has not been correctly collecting the intensity of the traffic. The optical sensor of the informational radar in its initial position either was not well calibrated or, due to the trees that were in its trajectory, did not adequately pick up the traffic in Segarra Street. Regarding the 400 vehicles per day that this sensor collected on average and that validated the hypothesis of the Barcelona City Council about a successful pacification of the commercial axis and nearby streets, it has been demonstrated with direct qualitative observation that the average number of vehicles at this point is 220 per hour on a working day.

Relating with the RQ2, if we consider that the measurements made reflect a constant volume from “7.00 h” to “22.00 h” (15 h), we can establish a real average of 3300 vehicles per day, recalling the complexity of access in the initial turn to this street, which is a street without commercial activity, with a single pylon and a 1 m sidewalk, passing through a square with a children’s playground. In Table 13, we can see an overview of all data collected at P2 (the most critical point in the new traffic loop):

As shown in Section 4.1, Section 4.2, Section 4.3 and Section 4.4 these traffic data (averaged from 10 m point samples at different times of the day) are corroborated by the study finally carried out by the Barcelona City Council (Section 4.5), based on the constant complaints of the “FemCrida” platform. The weekly average obtained, including weekends, establishes the flow at this point at approximately 3500 vehicles, a totally disproportionate figure for the type of street, housing and services.

The data obtained qualitatively in our study were corroborated by the mechanical sensors of the Barcelona City Council study, in that more than 70% of the vehicles going up Agustí i Milà Street turn onto Segarra Street to access the center of the neighborhood due to the change of direction resulting from the pacification of Gran Street. This is demonstrated both by our study and by the data obtained from the mechanical sensors, correctly interpreted. Importantly, the interpretation must be correct for proper decision making, because as discussed in Section 4.5, the analytical process presented by the City Council of Barcelona from the data they collected is not correct and has serious deficiencies. 

As shown, the qualitative approach used is validated by the actual data obtained by mechanical sensors, and even from the 70% interpretation indicated by the City Council. In other words, a minimum of five traffic samples of 10 min spread throughout the day at representative times have the same validity as quantitative data averaged by sensors over a week, with the extra benefit of being able to separate and identify the type of vehicles and other urban aspects such as speed, violations, turning problems, etc.

Approximately 3000 vehicles make the complete alternative route due to the pacification of the commercial axis, with the destination of their displacement being the same axis. The conclusion is that the change of direction has led to diverting this volume of traffic from a wide street with stores (the destinations of commercial traffic), residential buildings of up to seven floors, and multiple parking lots to single platform streets paved with pavement (in some cases) with no significant commercial activity (an average of one trade per 25/50 m) and a housing type of primarily single-family houses of two floors and residential buildings of four–five floors. This conclusion is drawn on the basis of direct samples, since the optical sensor of the informational radar does not detect it reliably, and due to the placement of the mechanical bands, the data collected cannot adequately support reasoning.

Therefore, and related with the RQ3, we can confirm that a return to the previous direction of the section studied about the main street would allow the volume indicated to be redirected to alternative streets. The visual sample presented has the added value of having identified and classified the typology of the circulating vehicles. This aspect has not been tracked by any of the sensors used by the City Council, so it has not been clarified how the volume of commercial traffic is especially detrimental to the area.

The commercial type of vehicle is the main cause of the deterioration of the whole area, with many samples of shattered pavement, broken and/or twisted pylons, deterioration of trees (one felled by impacts and another in the process of falling) and installation cabinets, etc., all of which are clear problems that affect the pacification of the area and put passers-by in daily and constant danger, as we can see in Figure 11. In addition, air and noise pollution have notably increased, making it clear that pacification has not been carried out correctly.

Finally, and related with RQ4, we have shown how qualitative sampling allows us to identify the typology of traffic, beyond the number of vehicles, this aspect being a differential fact in the study of problems and sustainability of the urban environment and its traffic. In addition, and due to the typology of the neighborhood (single platform, many of them with pedestrian preference, and mostly narrow and with very tight turns), the qualitative approach complements and provides significant information with respect to the quantitative one by identifying the direct cause of the deterioration of the pavement, street furniture, trees, etc. The samples carried out have identified the numerous medium and large commercial vehicles, which, due to the change of direction of the streets, have to make a route not adapted to their needs to arrive to the main commercial street. In this sense, the categorization of traffic is fundamental since it should allow its redistribution, generating more sustainable neighborhoods thanks in a large part to citizen participation. To this end, and thanks to the method and results obtained, it is clear that the qualitative approach reliably reflects reality, with an approximation to the overall quantity without significant differences. 

However, important limitations of the data collected are also identified, since they are related to the affected population and may prevent their use [99,100]. Humans affected by a situation can be highly emotional, which can impair their judgment. Questioning quality and being able to compare and/or corroborate it with objective data from sensors is vital to the scientific validity of any study and its subsequent applicability in other environments both spatially and temporally. The idea that citizens can be useful and effective sources of scientifically rigorous observations has a long history and requires the establishment of rigorous and replicable methods. In this sense, data quality has many aspects. The general definition of quality as “the totality of characteristics of a product that influence its ability to satisfy stated and implied needs” (ISO 2002) must be broken down into more detailed quality elements to be useful. ISO 19113 proposes the following data quality elements (ISO 2002): completeness, logical consistency, positional accuracy, temporal accuracy and thematic accuracy. These elements constitute internal data quality, i.e., they focus on the characteristics of the data independently of its possible use, with a complementary concept appearing, which is external data quality, or “fitness for use”, which assesses the suitability of certain data for a specific task in a specific domain [101,102]. We highlight the following aspects: Availability: It is unknown how much, what and from where the information will be provided. Unlike a sensor network, the collection of information from the population cannot be planned in advance to obtain an optimal configuration of observations for the phenomenon of interest.Data quality: Unlike physical sensors, human sensors cannot be calibrated and do not meet standards. In general, they are not trained to make specific observations.And finally, the critical question of data ownership. Should the data be owned by the data producers? Furthermore, if the data are analyzed to produce additional layers of information, who is responsible if decisions based on this information are wrong due to the lack of quality of the underlying data?

In conclusion, the issues of availability, quality, privacy, data ownership, accessibility, integrity and accountability must be thoroughly addressed all at once and not separately. In this context, a mixed approach that corroborates the data obtained from citizen participation by the sensor network appears as an optimal position within the context of smart governance.

To solve these complexities, a mixed or hybrid model should be considered, incorporating human-derived information, thus providing an additional perspective on the future development of smart cities. 

On the other hand, the great risk in any qualitative study is that the sample is not sufficient to obtain significant data with respect to the universe of the same. In our study and initially, an extensive qualitative sample was designed at points P1, P3 and P4, with the aim of comparing them with the quantitative sample of the pedagogical radar, and especially with the sample of the mechanical sensors M1–M5. Complementarily and in parallel, the measurement point P2 was activated. In this case, with only five samples distributed throughout the day and in different slots, it has been shown that these data are consistent with the data obtained at P1, P3 and P4 (qualitative), and to the quantitative samples (of M1–M5). Logically, as in any qualitative study (even quantitative ones), the entire possible universe is not always sampled. This aspect should not be a problem, and although it may detract from the study’s credibility due to lack of data, comparison and extrapolation between types of studies should be prevalent. Possible variations between study slots, hours, or days, or even the non-taking of nocturnal samples in our case, are not identified as a problem of accuracy. If the number of samples, their distribution throughout the day and their weighting is carried out in a systematic way (as we have done in our study), the comparison of data and their validity supports the method used. As can be seen from the comparison made in Table 13, the approximation made is correct, yields equivalent data between the different measurements, and therefore, and in response to RQ4, the qualitative approximation made is not only correct, but also characterizes the traffic typology in detail and appears to be a more useful and equally valid instrument than the data obtained quantitatively to make changes in the distribution of traffic in the area studied, and useful for any other area subject to be addressed in future studies. 

## 6. Conclusions

Recent studies have highlighted the importance of context in general, and spatiotemporal context in particular, in studies evaluating new smart city models and their interactions with humans [103]. Such work has found a large number of technologies and applications using in situ and mobile sensors in the context of smart cities, and a surprisingly limited use of remote sensing and citizen engagement approaches. Furthermore, it is observed that the use of a larger number of sensors does not necessarily translate into an improvement in the quality of life of smart city citizens. Remote sensing seems predestined to unravel the complexities of urban landscapes. However, the variety of spatial and temporal sizes of urban features and the resulting range of scales, as well as the fact that remote sensing essentially provides a “bird’s eye view”, make this approach somewhat unfavorable in the context of smart cities. For example, more users are adopting route recommendation systems. Typically, these systems look at historical and current traffic conditions to evaluate and recommend the fastest routes. However, in addition to mobility aspects, more contextual information, such as unplanned street events, dangerous locations, restrictions, and neighborhood safety, are not taken into account in the recommendation process [104].

This paper examines the case of the mobility changes that have occurred in the Sant Andreu District, Barcelona, Catalonia, after the pacification of a commercial main street, and compares the data obtained from sensors with the data observed by citizens. 

The contributions of this paper are:Findings: Smart cities research needs the contribution of the citizens.
○It has been shown that the placement of speed sensors is essential for proper data collection. Possible interferences with street furniture and trees distort the results, leading to incorrect decisions.○It has also been shown that the placement of mechanical sensors does reliably reflect the volume of traffic on the monitored streets, but that at the same time, this can be obtained through qualitative sampling in limited slots that cover the main periods of the day. ○Finally, it has been demonstrated that the qualitative sample obtained from citizen participation complements the quantitative one in a determining factor: the typology of vehicles and their relationship with the variables of the environment (time of transit, speed, turning problems, infractions, etc.), this being a very relevant datum since it can allow for an intelligent distribution of traffic that would not be possible from the data obtained by the sensors. Practical Implications: By identifying the potential for collaboration between citizens motivated to improve their neighborhood and proposals based on smart city monitoring promoted by administrative entities, a connection that has so far been poorly represented in the literature is generated. This article encourages a more structured discussion between academia (and systematic data collection) and policy makers focused on the sustainable development of cities/urban areas. Based on the results shown in our study, a bilateral communication process has been initiated between the City Council and the FemCrida platform. As a result of this process, problems have been identified and a timetable for improvement actions, previously non-existent, has been established, demonstrating that the approach taken and justified in this article generates new synergies in the improvement of the city.Social Implications: This paper describes how citizens can serve as human sensors to provide supplementary, alternative and complementary sources of information for smart cities and sensors. Our paper highlights citizen participation as an important part of the smart city concept, where citizens participate to help the city solve its problems. The creation of a participatory innovation ecosystem where citizens and communities interact with public authorities and knowledge developers is key. This collaborative interaction results in co-designed, user-centered innovation services and calls for new governance models. Urban transformation in which citizens are the main “drivers of change” through their empowerment and motivation ensures that major city challenges, including sustainable behavioral transformations, can be addressed.Research limitations/implications: The paper highlights the need to explore the question of how specific contexts in which particular urban areas are located influence the mobility and development strategies of those areas. Further research is needed to advance mixed studies that characterize urban mobility and possible changes for effective and real pacification of specific areas. In this sense, the challenges of the city can be more effectively addressed on the neighborhood scale and both our study and others conducted offer examples and experiences that demonstrate the feasibility, importance and impact of such an approach [28].Originality/value: Our paper describes the relationship between urban data collection processes and changes in the mobility design of new smart cities. By characterizing the data sources, it identifies possible biases and allows us to establish new functional connections between the set of ideas about what new, more peaceful and smart cities should be and what is feasible to implement due to the current commercial and housing model.

Smart cities are concerned with increasing the quality of life, providing better services, reducing the environmental footprint and improving citizen participation [76]. When citizens act as sensors, they contribute to all these aims of Smart Cities. City services become more effective since the city is alerted to problems, including environmental issues. Citizens participate to improve the city and thereby the quality of life, in a process clearly identified as Smart Governance. Future research may include further analysis of the data we have obtained. Another possibility is to investigate the motivation of citizens who contribute as human sensors. 

With our study, we have demonstrated how a poor placement and/or calibration of a sensor can yield incorrect data leading to inappropriate decisions, which seriously affect citizens. Moreover, with data corroborated by the mechanical sensors installed, we have shown how a qualitative sample based on temporary sampling throughout the day can obtain data comparable to a continuous sample, with the addition of having parameterized vehicle types, directions of movement, violations, etc., something that exceeds the data collected mechanically and/or digitally. In short, a qualitative and participatory sample makes it possible to obtain reliable, usable and verifiable data that will enable decision making for urban redevelopment with greater consensus among all stakeholders in today’s cities. Only the distribution of vehicles according to their needs throughout the environment and/or the limitation of entry to the neighborhood by vehicle type, schedule, or proximity appear to be possible solutions. Given the typology of the neighborhood, medium and large commercial vehicles bring serious complications and generate constant damage. A final distribution of smaller vehicles or distribution centers bordering the neighborhood where the residents should go, together with certain complementary actions of grouped distribution by hour, etc., is seen as having a real capacity to reduce traffic and to be able to show that pacification has been carried out correctly.

Based on the data collected and shared with the City Council, the latter has changed the location of the optical sensor of the informational radar in the same street. The current, more delayed position allows for a better field of view, and as observed in the visual measurements that are being carried out in a second phase, the detection effectiveness has increased from 85% to 98–99%. Likewise, with the preliminary data collected, there is still a greater increase in the average speed, which places all types of vehicles above the limit established for this street, and some kind of action is necessary to modify this situation. Finally, as we have introduced, a new round of data collection is being carried out with the aim of comparing the results with the optical sensor samples from 9 May 2021, the date on which the state of Alarm in Spain was lifted, limiting commercial activity and establishing a curfew from “22:00 h” to “06:00 h”. The first data collected, pending validation and analysis, reflect an increase of between 20 and 30% in all the strips, which puts the volume of entry at approximately 6000 vehicles per day, which would correlate with a volume in the conflict zone studied of between 4000 and 4500 vehicles per day.

Finally, it is clear that the design of the activities and the urbanism of today’s cities requires the participation of all the actors involved. The Game4City project, which is the focus of this research, demonstrated in its previous iterations [105,106,107,108,109] that by visualizing and interacting with future designs, both citizens and the entities that promote them can make better decisions. Simulation with interactive virtual reality systems of specific areas for specific uses allows for the co-design of space and its functions, democratizing the cities of the future [105,110]. Decisions based on sensors provide biased information, either by calibration or by lack of details, such as the type of vehicle in our case. Identifying the mobility required for each vehicle and person should make it possible to personalize routes, distribute traffic and effectively pacify our streets, neighborhoods and, therefore, cities, which is a perspective that government entities should consider. With the proposals that can be developed jointly by neighborhood organizations, educational areas and government teams, it is easier, cheaper and more sustainable to reach a meeting point, but this also requires political will.

## Figures and Tables

**Figure 1 sensors-21-05321-f001:**
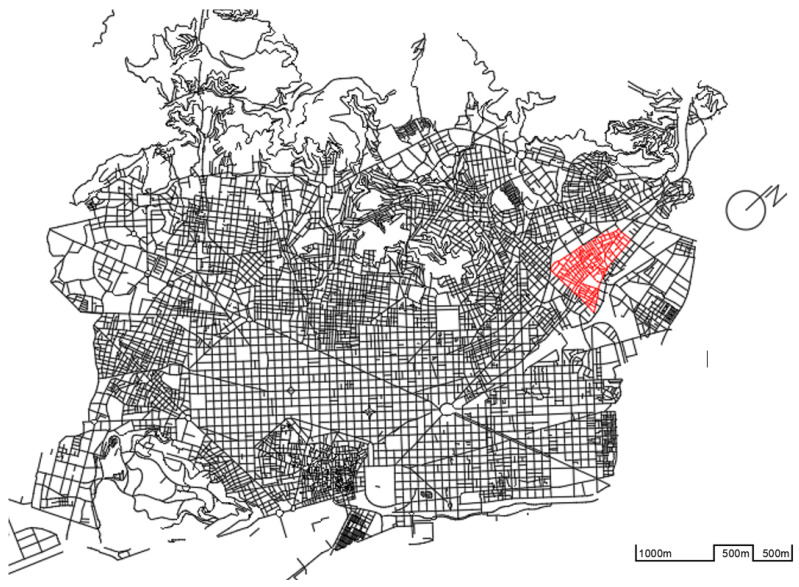
Location of Sant Andreu del Palomar in Barcelona City.

**Figure 2 sensors-21-05321-f002:**
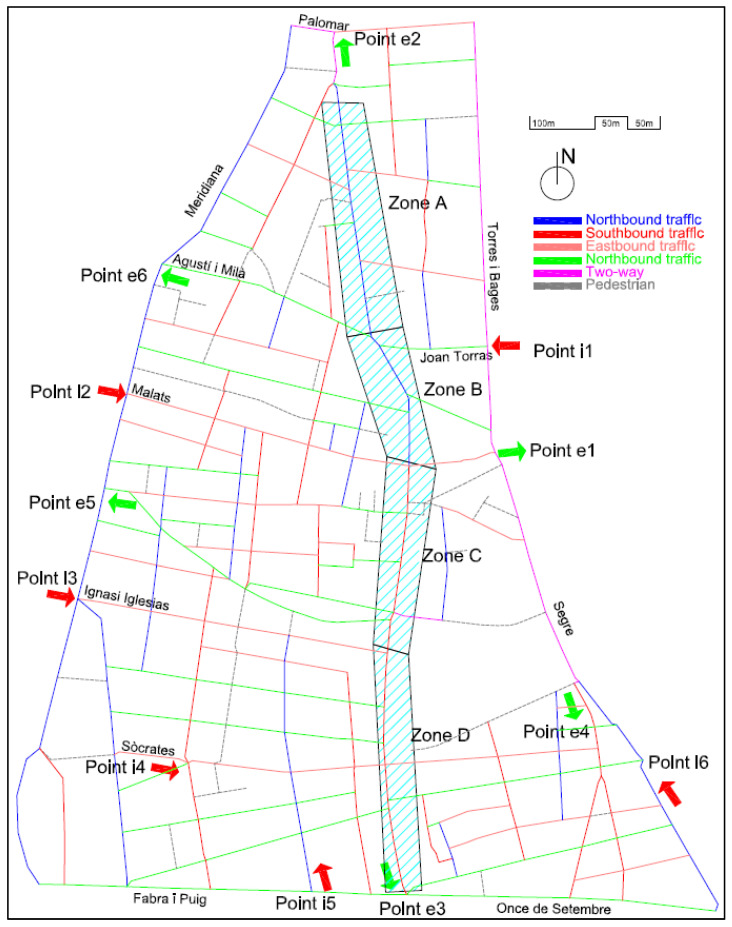
Historic center of the Sant Andreu del Palomar neighborhood.

**Figure 3 sensors-21-05321-f003:**
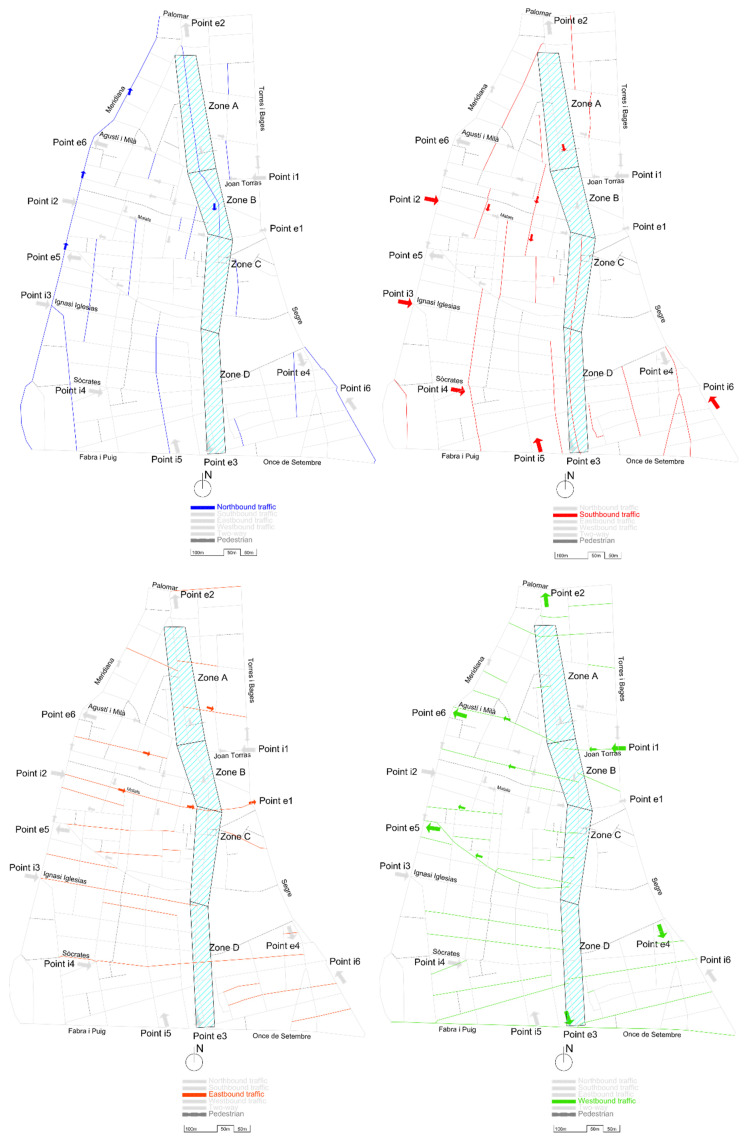
Direction of the traffic by street after the pacification of Gran Street.

**Figure 4 sensors-21-05321-f004:**
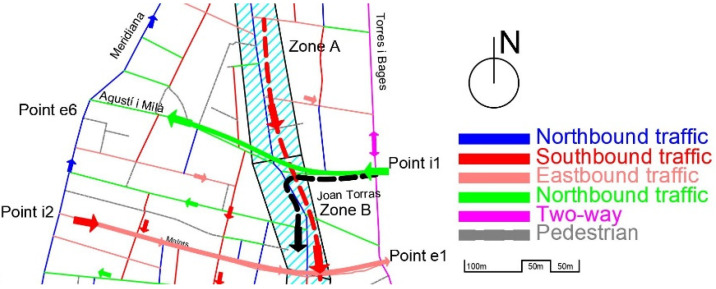
Initial street configuration before the pacification.

**Figure 5 sensors-21-05321-f005:**
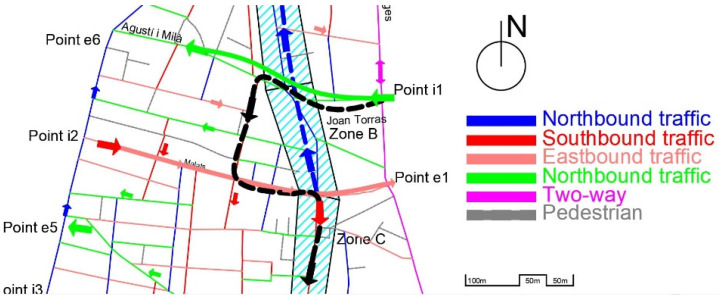
Street configuration after the pacification, where in black we can see the route to arrive at the center of the neighborhood.

**Figure 6 sensors-21-05321-f006:**
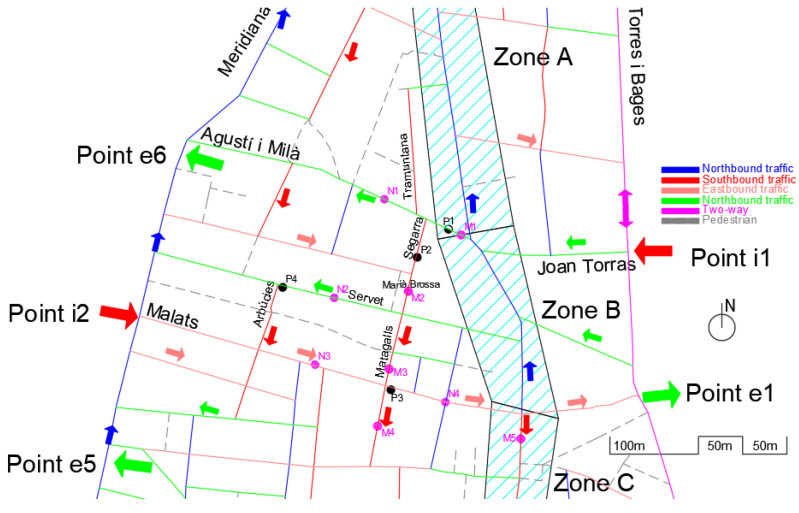
Area of study, with the locations of the four measurement points.

**Figure 7 sensors-21-05321-f007:**
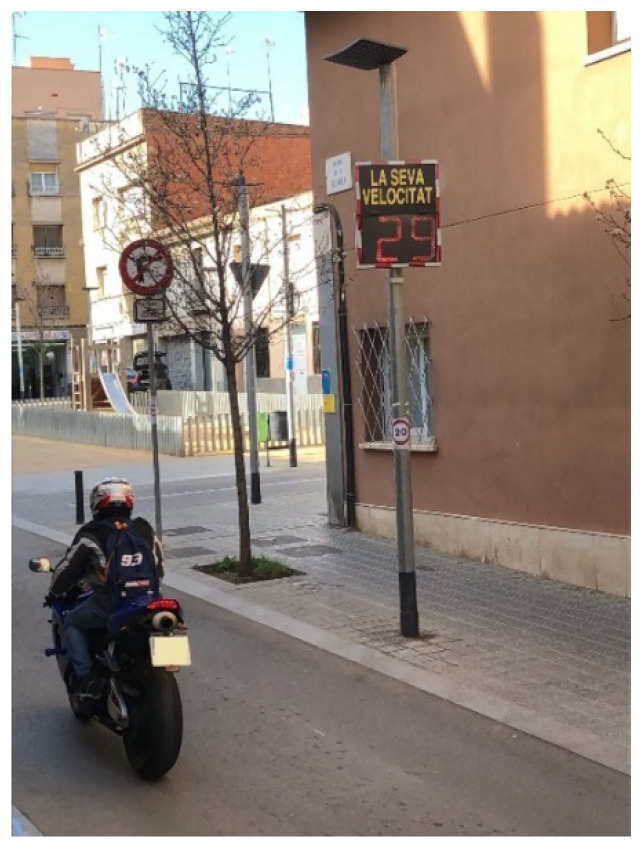
Informational radar on Segarra St. (photo by author), located at P2 (see Figure 6).

**Figure 8 sensors-21-05321-f008:**
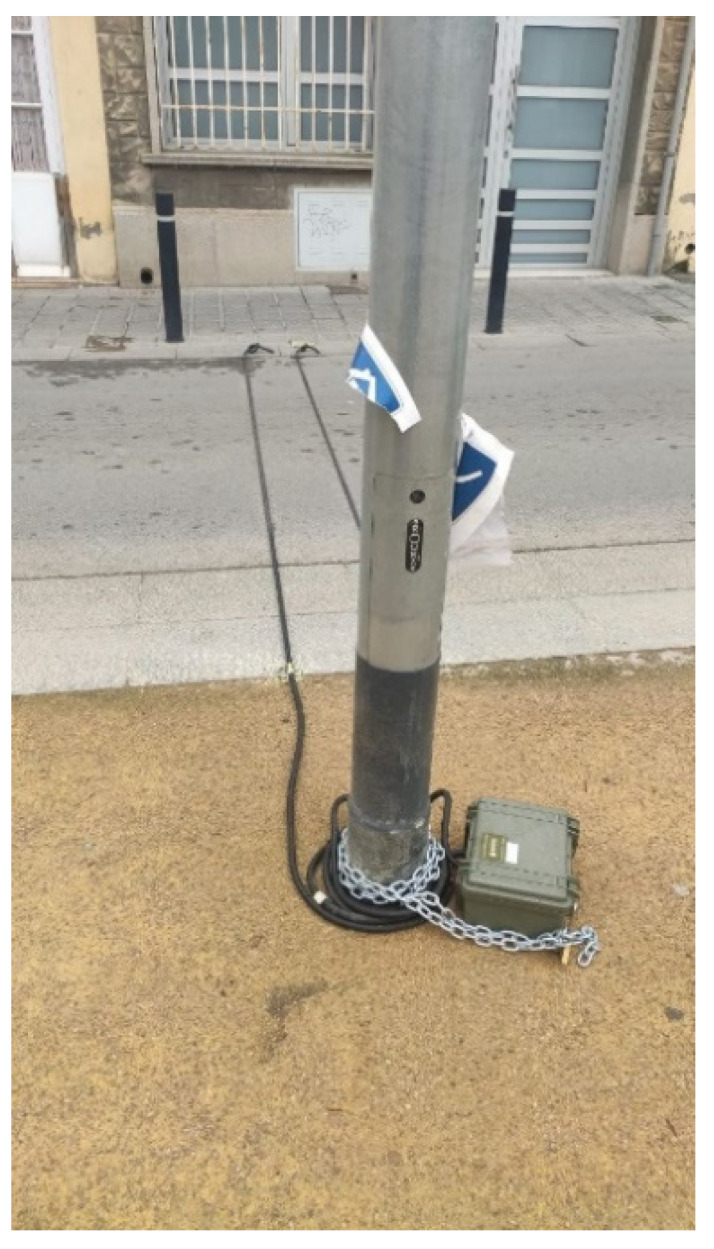
Circulation counting mechanical sensor (photo by author) located at M2 according to Figure 6.

**Figure 9 sensors-21-05321-f009:**
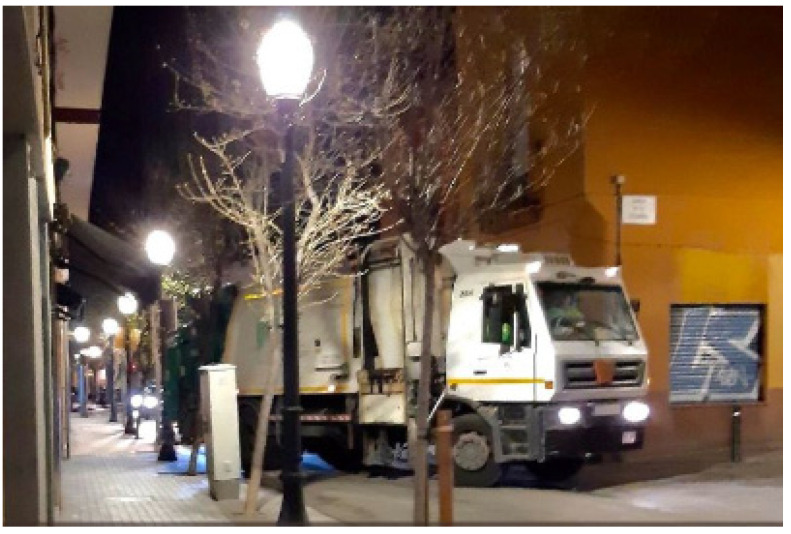
Example of a CDP for a vehicle in circulation between P1 and P2.

**Figure 10 sensors-21-05321-f010:**
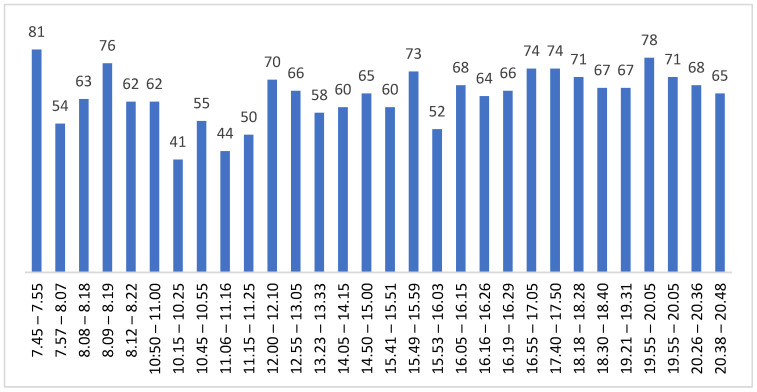
Time distribution on working days.

**Figure 11 sensors-21-05321-f011:**
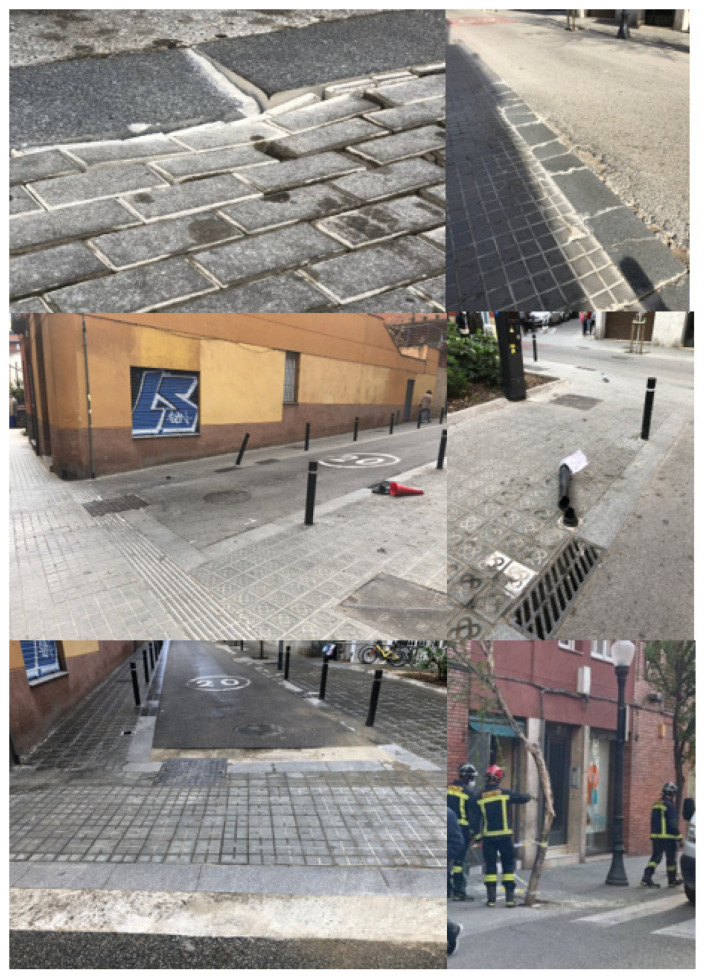
Some examples of the problems referenced in the main text between the P1 and P2 measurement points.

**Table 1 sensors-21-05321-t001:** Morning sample resume on business days.

Sample N°	Morning (7:00–13:00)	Total	V1	V2	V3	V4	CDPs
	Total (2 h, 12 samples)	724	116	348	109	151	67
	Veh/Hour	362	58	174	54.5	75.5	33.5
	%		16.02	48.07	15.06	20.86	9.25
	Average (10 min)	60.33	9.67	29.00	9.08	12.58	5.58
	Standard Deviation (SD—10 m)	12.16	4.01	9.51	2.61	3.90	2.81

**Table 2 sensors-21-05321-t002:** Noon sample resume on business days.

Sample N°	Noon (13:00–16:00)	Total	V1	V2	V3	V4	CDPs
	1 h (6 samples)	368	64	209	41	54	26
	% sobre total HORA		17.39	56.79	11.14	14.67	7.07
	Average	61.33	10.67	34.83	6.83	9.00	4.33
	SD (10 m)	7.09	4.32	7.05	1.94	4.77	2.34

**Table 3 sensors-21-05321-t003:** Afternoon sample resume on business days.

Sample N°	Afternoon (16:00–21:00)	Total	V1	V2	V3	V4	CDPs
	Total (2 h, 12 samples)	833	170	541	61	61	41
	Veh/Hour	416.5	85	270.5	30.5	30.5	20.5
	%		20.41	64.95	7.32	7.32	4.92
	Av. (10 min)	69.42	14.17	45.08	5.08	5.08	3.42
	(SD—10 m)	4.23	3.13	4.85	4.32	2.91	1.78

**Table 4 sensors-21-05321-t004:** Saturday sample resume.

Sample N°	Saturday	Total	V1	V2	V3	V4	CDPs
	Total (2 h, 12 samples)	499	94	340	30	35	21
	Average for 1 h	249.5	47	170	15	17.5	10.5
	% by hour		18.84	68.14	6.01	7.01	4.21
	Average (10 m)	41.58	7.83	28.33	2.50	2.92	1.75
	SD (10 m)	9.30	3.61	5.76	2.11	1.83	1.29

**Table 5 sensors-21-05321-t005:** Sunday sample resume.

Sample N°	Sunday	Total	V1	V2	V3	V4	CDPs
	Total (2 h, 12 samples)	334	52	249	16	17	9
	Average for 1 h	167	26	124.5	8	8.5	4.5
	% by hour		15.57	74.55	4.79	5.09	2.69
	Average (10 m)	27.83	4.33	20.75	1.33	1.42	0.75
	SD (10 m)	12.46	3.42	9.45	1.44	1.00	0.75

**Table 6 sensors-21-05321-t006:** Qualitative sample for Segarra Street.

	Enter Segarra	Radar Vel. Av.	Radar Vel. Max.	Follow on Agustí St.	TOTAL (eq. to P1)
“9:35 h–9:45 h”					
V1	5	24.3	35	1	
V2	15	21.4	27	5	
V3	8	21.5	27	0	
V4	12	21.4	27	5	
TOTAL	40			11	51
Radar does not work	5				
From Tramuntana St.	3				
CDPs	7				
“9:47 h–9:57 h”					
V1	4	31	36	6	
V2	15	23	32	10	
V3	1	31	31	2	
V4	3	23.8	26	1	
TOTAL	23			19	42
Radar does not work	2				
From Tramuntana St.	1				
CDPs	2				
“11:34 h–11:44 h”					
V1	6	19.1	22	4	
V2	13	18.4	22	5	
V3	10	18.5	20	3	
V4	5	18.3	23	3	
TOTAL	34			15	49
Radar does not work	6				
From Tramuntana St.	4				
CDPs	3				
“16:48 h–16:58 h”					
V1	7	19.9	22	6	
V2	20	18.2	25	17	
V3	4	19.5	20	0	
V4	6	18.5	25	4	
TOTAL	37			27	64
Radar does not work	5				
From Tramuntana St.	3				
CDPs	4				
“17:06 h–17:16 h”					
V1	5	22	22	7	
V2	27	18.9	27	19	
V3	7	21.3	22	0	
V4	10	19.7	22	1	
TOTAL	49			27	76
Radar does not work	10				
From Tramuntana St.	2				
CDPs	6				

**Table 7 sensors-21-05321-t007:** Qualitative sample summary for Segarra Street by type of vehicle.

Type	C/Segarra	%	Velocity (Av.)
V1	27	14.75	23.26 km/h
V2	90	49.18	19.98 km/h
V3	30	16.39	22.36 km/h
V4	36	19.67	20.34 km/h
TOTAL (Segarra Street)	183		
TOTAL (Agustí i Milà Street)	99		
Radar does not work	28	15.3	
From Tramuntana Street	13	7.1	
CDPs	22	12.0	
Excess of velocity	61	33.3	
Average (10 m):	36.6	Veh.	
Average (1 h):	219.6	Veh.	

**Table 8 sensors-21-05321-t008:** Matagalls Street sample resume, square with Malats Street.

Sample	Total	V1	V2	V3	V4
8 Samples (10 m)	274	54	125	72	23
SD (10 m)	7.79	3.15	2.26	2.32	1.30
Average by hour	205.5	40.5	93.75	54	17.25
%		19.71	45.62	26.28	8.39

**Table 9 sensors-21-05321-t009:** Servet Street corner and Arbúcies Street, morning sample resume.

Date	Time	Total	V1	V2	V3	V4
	Av. (10 m)	13.47	3.25	8.29	1.5	0.82
	SD (10 m)	4.65	1.48	4.24	1.15	1.01
	Av. by hour	80.82	19.5	49.74	9	4.92
	%		24.13	61.54	11.14	6.09

**Table 10 sensors-21-05321-t010:** Servet Street corner and Arbúcies Street, noon sample resume.

Date	Time	Total	V1	V2	V3	V4
	Av. (10 m)	14.0	2.5	8.9	2.2	0.6
	SD (10 m)	4.44	1.73	3.96	1.53	0.9
	Av. by hour	84	15	53.52	13.02	3.48
	%		17.86	63.71	15.50	4.14

**Table 11 sensors-21-05321-t011:** Servet Street corner and Arbúcies Street, afternoon sample.

Date	Time	Total	V1	V2	V3	V4
	Av. (10 m)	16.81	2.81	11.62	1.52	0.86
	SD (10 m)	5.81	2.27	3.79	1.72	1.28
	Av. by hour	100.86	16.86	69.72	9.12	5.16
	%		16.71	69.12	9.04	5.11

**Table 12 sensors-21-05321-t012:** Servet Street global data resume.

Number of Samples	Hours	Total	V1	V2	V3	V4
56	9.33	353	59	244	32	18
	% by hour		18.85	64.47	11.28	5.40
Total by hour		89.28	16.83	57.56	10.08	4.82

**Table 13 sensors-21-05321-t013:** P2, Segarra street data comparison. Vehicles by day.

Optical Sensor Average	P2 Direct Data Collection	M2 Mechanical Sensor Data	P1 Estimation (60–70%)
430.25	3294	3537	3465–4042
Obtained by average of eight days data collection	Obtained by average of five direct citizen collection	Obtained by average of five direct mechanical data collection	Obtained by average of 30 direct citizen collection, using samples of 10 min

## Data Availability

Publicly available datasets were analyzed in this study. This data can be found here: https://drive.google.com/file/d/1MSXVIKocyKRFjPXYyuQ3-PTNAQvjaA_H/view?usp=sharing.

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
