# Peer review of "Towards Smart City Governance. Case Study: Improving the Interpretation of Quantitative Traffic Measurement Data through Citizen Participation"

_sensors, 2021, doi:10.3390/s21165321_

Round 1

Reviewer 1 Report

The title requires reflection and consideration. It is neither scientific nor encouraging to read. It needs to be revised.  It should be made more scientific and relate to the content of the article.  
The thesis is not clear, there are no research questions. 
The term "Citizens as Sensors" is not justified and legitimate. It should be corrected. 
The literature analysis could be developed more towards the title (only that the title should be changed)
The methodology is not justified, and posting pictures is not scientific if you do not substantively justify the methodology. Parts are disconnected from each other. There is no linkage. A smooth transition. The article should be structured differently. Use the good data that the authors have and add a good theoretical underpinning, and to this add a very good scientific title 

Author Response

Dear Reviewer,

thank you so much for your time and efforts. We have improved the manuscript following your suggestions as well as the other comments received by other reviewers. We hope that you can enjoy the new version.  You can view the detailed answers in the document attached.

Best Regards.

Reviewer 2 Report

The paper discussed the effectiveness of sensor data in urban decision making and compared it with the data obtained by means of qualitative observation in a specific scenario of city pacification of a small town in Barcelona city. It has some reference value for data collection and analysis in urban decision making. However, there are many deficiencies in the paper.
1. The key idea and main contribution of the paper is confusing. The authors gives out too many key words. Which approach is better, the sensor data collection or qualitative observations? In which aspect, it is better? Why?
2. The structure of the paper need to be reorganized. Aside from the background, the key concept and theory can be discussed in a separate Section, instead of being the subsection of introduction. The details of the scenario should be described in the material section, not the introduction section. In the mean while, the general discussion of urban problem need to be placed at the beginning of the paper, not in the material section. The method section need to be extended. It is necessary to add a discussion section after the result.
3. Please add scale lable to all of the maps. Many maps missed scale lable.
4. The presentation of data need to be improved. It is suggested to present the results of sensor data in forms, instead of just with words. There are too many data forms for qualitative observation. It is suggested to merge them together, highlighting the key ideas of the paper. Why not the authors directly compare the sensor data with the observation data in some form?
5. The recorded data need to be presented in some order in the tables, such as the ascending time.
6. the study scenario of the paper is hard to understand. For better explanation, it is suggested to lable the direction of vehicle curculation in the maps. Why the authors chose P1, P2, P3, and P4 for qualitative observation?
7. What is the relationship between the data of M1, M2, M3, M4 and the data of P1, P2, P3, P4?
8. Some content in the conclusion part should be introduced at the beginning of the paper, e.g. the concept of city pacification. The conclusion part should be focus on the contribution of the paper and outlooks.

Author Response

(The authors gave the same response as above.)

Round 2

Reviewer 1 Report

Previous comments have been addressed, but I still do not know what benefit and contribution this article makes to science. It is a review article, I do not know what knowledge other scientists can use and how. As a case study it is ok, but there is no strong scientific contribution and this does not convince me to publish it. There is no indication that similar solutions are used in other countries, other cities, or that it is so unique and innovative that others should use it. This is not apparent from an analysis of the text. 

Author Response

Dear Reviewer,

thanks for your review, time and efforts, as well as your critical thinking. We have done a thorough review and contextualization of the study to relate it and base it in a scientific way without any doubt. Apart from the completeness of the study, the new contextualization of the study and its relationship with citizen science and smart governance, we believe that it provides an adequate response to your requirements.
We sincerely hope that you can appreciate our efforts and changes made to finally obtain a positive response. 

Kindly.

Best Regards

Reviewer 2 Report

It's improved after revision, but I don't think it is qualified to be publish in current state.
1. The authors mentioned the participants role of citizens in the sustainable development of cities many times in Abstract and main content of the paper. However, the idea is not refelcted by the title of the paper. So, in essence, what is the key idea of the paper?
2. The authors should not put too much details of the study area in the Introduction section, the details should be in the Material section. Figure 1 should be moved to Section 3.
3. There are two Section 2 in the revised version of the paper. Please check and make modification to the paper carefully.
4. As for any of Figure 2, 3, and 7, the authors merged different types of traffic flow in a single picture. Although directions are labelled now, it is still hard to understand the traffic patterns. For better explanation, it is suggested the authors to separate each figure into several ones, which highlight different types of traffic flow in different pictures respectively.
5. The theoretical disucssion of the paper is limited. Why qualitative observation can improve quantitative sensor's data? I think it is because it is more detailed and rich in content. However, quanlitative observation is not as accurate as the sensor's data, and it cannot record the state of cities in 24 hours of a day. So, in what stage it would be helpful? The authors should discuss the problem in a higher level, not just give a description of the experiment results.

Author Response

Dear reviewer,

thank you very much again for your time, efforts and suggestions to improve our paper. We believe we have addressed all your comments in a positive manner. 

We look forward to a positive response.

Best regards.
